

1          18 May 2018

# New and improved infrared absorption cross sections for trichlorofluoromethane (CFC-11)

11          by

13          Jeremy J. Harrison[1,2,3]

[1]*Department of Physics and Astronomy, University of Leicester, Leicester LE1 7RH, United*
*Kingdom.*
[2]*National Centre for Earth Observation, University of Leicester, Leicester LE1 7RH, United*
*Kingdom.*
[3]*Leicester Institute for Space and Earth Observation, University of Leicester, Leicester LE1*
*7RH, United Kingdom.*
Number of pages   = 18
Number of tables   = 3
Number of figures = 5


Address for correspondence:

Dr. Jeremy J. Harrison

National Centre for Earth Observation

Department of Physics and Astronomy

University of Leicester

University Road

Leicester LE1 7RH

United Kingdom


*e-mail*:          jh592@leicester.ac.uk



**Abstract**
Trichlorofluoromethane (CFC-11), a widely used refrigerant throughout much of the
twentieth century and a very potent (stratospheric) ozone depleting substance (ODS), is now
banned under the Montreal Protocol. With a long atmospheric lifetime, it will only slowly
degrade in the atmosphere, so monitoring its vertical concentration profile using infrared-
sounding instruments, thereby validating stratospheric loss rates in atmospheric models, is of
great importance; this in turn requires high quality laboratory spectroscopic data.
This work describes new high-resolution infrared absorption cross sections of
trichlorofluoromethane / dry synthetic air over the spectral range 710 – 1290 cm$^{-1}$,
determined from spectra recorded using a high-resolution Fourier transform spectrometer
(Bruker IFS 125HR) and a 26-cm-pathlength cell. Spectra were recorded at resolutions
between 0.01 and 0.03 cm$^{-1}$ (calculated as 0.9/MOPD; MOPD = maximum optical path
difference) over a range of temperatures and pressures (7.5 – 760 Torr and 192 – 293 K)
appropriate for atmospheric conditions. This new cross-section dataset improves upon the
one currently available in the HITRAN and GEISA databases.





## 1. Introduction

Chlorofluorocarbons (CFCs) were first developed in the 1930s as safe, reliable, and
non-toxic refrigerants for domestic use.  Trichlorofluoromethane, known as CFC-11 or
Freon-11, and dichlorodifluoromethane, known as CFC-12 or Freon-12, were the two most
widely used CFCs in applications ranging from refrigerators and air conditioners to
propellants in spray cans and blowing agents in foam production.
Ultimately, however, CFCs proved too good to be true.  The explosion in their use
led to a steady increase in their atmospheric abundances.  While inert in the troposphere, it
was this stability which enabled them to reach the stratosphere where dissociation by
ultraviolet radiation released chlorine atoms, which catalyse the destruction of stratospheric
ozone.  The realisation of this impending environmental disaster prompted international
action and in 1987 the Montreal Protocol was ratified; this led to the phasing out of the
worldwide production and use of CFCs.  CFCs are still released into the atmosphere from
"banks", such as old refrigerators, however these are not regulated by the Protocol (Harris et
al., 2014).  Banks are the major source of emissions for many ODSs, including CFC-11
which has a long atmospheric lifetime of 52 years (Harris et al., 2014).
At present, CFC-11 is the second most abundant CFC in the atmosphere and
contributes the second-highest amount of chlorine to the stratosphere, behind CFC-12.  In
addition to its role in stratospheric ozone destruction – it has the highest ozone depletion
potential (1.0) (Harris et al., 2014) of all the CFCs – CFC-11 is a particularly strong
greenhouse gas – it has a 100-yr global warming potential of 5160 (Harris et al., 2014).
As a key species in stratospheric ozone destruction, CFC-11 atmospheric
concentrations are monitored in situ at the surface, e.g. the annual global mean mole fraction
of CFC-11 measured by the AGAGE (Advanced Global Atmospheric Gases Experiment)
network in 2012 was 235.5 ppt (Carpenter et al., 2014).  However, in order to measure
concentrations in the stratosphere where ozone destruction occurs, remote-sensing techniques
are required.  Table 1 contains a listing of limb sounders capable of measuring CFC-11, as
described in the literature.
The infrared (IR) spectra for large molecules like trichlorofluoromethane are highly
complex, consisting of very many closely spaced spectroscopic lines, making the task of
generating line parameters from measurements an almost impossible one.  For the purposes
of atmospheric remote sensing, it is possible to use absorption cross sections in forward
models instead of line parameters, however this requires laboratory measurements of air-
broadened spectra over a range of temperatures and pressures.  The accuracy of retrievals of
CFC-11 abundances for the limb sounders in Table 1 is very much dependent on the quality
of the underlying spectroscopy; ideally absorption cross-section datasets should cover a range
of atmospherically relevant pressure-temperature (PT) combinations, with accurate
wavenumber scales and band intensities, and properly resolved spectral features. This work
presents new spectroscopic data, optimised for limb sounding instruments, which improve
upon those currently available in the HITRAN and GEISA databases.

**2. Infrared spectroscopy of trichlorofluoromethane**
**2.1. Spectroscopic background**
There are two stable isotopes of carbon and chlorine, and one of fluorine, resulting
in eight stable isotopologues of trichlorofluoromethane, namely $^{12/13}C^{35}Cl_3F$, $^{12/13}C^{35}Cl_2^{37}ClF$,
$^{12/13}C^{35}Cl^{37}Cl_2F$, and $^{12/13}C^{37}Cl_3F$; these belong to the point groups $C_{3v}$, $C_s$, $C_s$ and $C_{3v}$,
respectively. Taking into account the natural abundances of $^{12}C$ / $^{13}C$ (~ 99% and ~1%), and
$^{35}Cl$ / $^{37}Cl$ (~ 76% and ~24%), the most abundant isotopologues are therefore $^{12}C^{35}Cl_3F$,
$^{12}C^{35}Cl_2^{37}ClF$, and $^{12}C^{35}Cl^{37}Cl_2F$, with abundances of 43%, 41%, and 13%, respectively.
As a non-linear molecule with five atoms, trichlorofluoromethane possesses nine
normal vibrational modes; in the $C_{3v}$ point group there are three non-degenerate
fundamentals of $A_1$ symmetry ($\nu_1$, $\nu_2$, and $\nu_3$), and three doubly-degenerate fundamentals of E
symmetry ($\nu_4$, $\nu_5$, and $\nu_6$). For the $C_s$ point group, the $\nu_1$, $\nu_2$, and $\nu_3$ modes possess A'
symmetry, with the doubly-degenerate $\nu_4$, $\nu_5$, and $\nu_6$ modes each splitting into one A' and one
A'' mode (Snels et al., 2001). Since the splittings in the $\nu_4$, $\nu_5$, and $\nu_6$ levels are small, it is
normal to label these bands assuming $C_{3v}$ symmetry. The 710 – 1290 cm$^{-1}$ spectral range
covered in the present work contains two strong fundamental bands, $\nu_1$ ~ 1081.28 cm$^{-1}$ and $\nu_4$
~ 849.5 cm$^{-1}$ , and a weaker combination band, $\nu_2 + \nu_5$ ~ 936.5 cm$^{-1}$; reported frequencies are
those for the most abundant isotopologue, $^{12}C^{35}Cl_3F$ (von Lilienfeld et al., 2007 ; Snels et al.,
2001). Isotopologues complicate the already dense $CCl_3F$ rotation-vibration spectrum; each
has slightly different molecular parameters, with bands shifted by small amounts relative to
each other. These main band systems are shown in Figure 1 in the plot of the new absorption
cross section at 191.7 K and 7.535 Torr. Details on the measurement conditions and
derivation of this cross section are given in Section 3.

**2.2. A brief history of trichlorofluoromethane absorption cross sections**
High resolution (0.03 cm$^{-1}$) absorption cross sections of pure trichlorofluoromethane
at 296 K were first included in HITRAN as part of the 1986 compilation (Massie et al., 1985;



Rothman et al., 1987). The HITRAN 1991/1992 compilation saw the first introduction of
temperature-dependent cross sections (203 – 293 K) for CFC-11 (McDaniel et al., 1991;
Rothman et al., 1992; Massie and Goldman, 1992); as before these were derived from
measurements of pure $CCl_3F$ at 0.03 cm$^{-1}$ resolution.

While the two previous HITRAN editions (1986 and 1991/1992) neglected pressure-

broadening effects on the $CCl_3F$ spectra, cross sections for 33 distinct PT combinations (201–
296 K and 40–760 Torr $N_2$-broadened) over two wavenumber ranges, 810–880 cm$^{-1}$ and
1050–1120 cm$^{-1}$, were introduced into HITRAN 1996 (Li and Varanasi, 1994; Rothman et
al., 1998). Another 22 PT combinations covering lower pressures and temperatures over the
same wavenumber ranges were added to HITRAN 2000 (provided by Varanasi, cited within
Rothman et al., 2003), bringing the overall PT coverage to 190−296 K and 8−760 Torr. Out
of these 55 PT combinations, four pairs possess both temperature and pressure within 1 K
and 5 Torr, respectively. This dataset, henceforth referred to as the Varanasi dataset, has
been used widely for remote-sensing applications since it was first introduced; it is still the
dataset included in the most recent GEISA 2015 (Jacquinet-Husson et al., 2016) and
HITRAN 2016 (Gordon et al., 2017) spectroscopic databases. Despite its widespread use,
the Varanasi dataset has some deficiencies which will be discussed in Section 4, alongside a
comparison with the new spectroscopic data taken as part of the present work.

**3. New absorption cross sections of air-broadened trichlorofluoromethane**
**3.1. Experimental**

The experimental setup at the Molecular Spectroscopy Facility (MSF), Rutherford

Appleton Laboratory (RAL) and the experimental procedures have been described previously
for related measurements (e.g. Harrison et al., 2010; Harrison, 2015b; Harrison, 2016); the
reader is referred to one of these previous studies for more information. Instrumental
parameters associated with the Fourier Transform Spectrometer (FTS) used for the
measurements, sample details, and the cell configuration are summarised in Table 2. The
sample pressures and temperatures for each air-broadened spectrum, along with their
experimental uncertainties and associated spectral resolutions, are listed in Table 3.

**3.2. Generation of absorption cross sections**

The procedure used to generate absorption cross sections from measured spectra has

been reported previously (e.g. Harrison et al., 2010; Harrison, 2015b; Harrison, 2016), so the
full details are not provided here. The wavenumber scale of the cross sections is calibrated
against the positions of isolated $N_2O$ absorption lines taken from the HITRAN 2012 database
(Rothman et al., 2013). The absorption cross sections, $\sigma(\upsilon, P_{air}, T)$ in units of cm$^2$ molecule$^-$
$^1$, at wavenumber $\upsilon$ (cm$^{-1}$), temperature $T$ (K) and synthetic air pressure $P_{air}$, are normalised
according to

$$\int_{710\,\mathrm{cm}^{-1}}^{1290\,\mathrm{cm}^{-1}} \sigma(\upsilon, P_{air}, T)\,\partial\upsilon = 9.9515 \times 10^{-17} \text{ cm molecule}^{-1}, \qquad (1)$$


where the value on the right hand side is the average integrated band intensity over
the spectral range 710 – 1290 cm$^{-1}$ for three 760-Torr-$N_2$-broadened trichlorofluoromethane
spectra (at 278, 298, and 323 K) from the Pacific Northwest National Laboratory (PNNL) IR
database (Sharpe et al., 2004). This intensity calibration procedure counters problems with
trichlorofluoromethane adsorption in the vacuum line and on the cell walls, and furthermore
assumes that the integrated intensity over each band system is independent of temperature.
The reader is referred to Harrison et al. (2010) for a more complete explanation of the
underlying assumption, and references cited within Harrison (2015a, 2015b, and 2016) for
details on previous successful uses of this approach.
A selection of the derived absorption cross sections is presented in Figure 2,
showing the expected behaviour with temperature at a total pressure of ~ 200 Torr; the
wavenumber range covers the microwindow for the ACE-FTS v3.6 retrieval scheme.

**3.3. Absorption cross section uncertainties**
The accuracy of the wavenumber scale for the new absorption cross sections is
comparable to the accuracy of the $N_2O$ lines used in the calibration; according to the
HITRAN error codes, this is between 0.001 and 0.0001 cm$^{-1}$. The uncertainty in the intensity
is dominated by systematic errors. A true measure of the random errors would ideally
require multiple concentration-pathlength burdens at each PT combination, however only one
is available for each. The maximum systematic uncertainties in the sample temperatures ($\mu_T$)
and total pressures ($\mu_P$) are 0.4 % and 0.7 %, respectively (see Table 3). The photometric
uncertainty ($\mu_{phot}$), which includes systematic error arising from the use of Bruker's non-
linearity correction for MCT detectors, is estimated to be ~2 %. The pathlength error ($\mu_{path}$)
is estimated to be negligibly small, lower than 0.1 %. According to the PNNL metadata, the
systematic error in the PNNL $CCl_3F$ spectra used for the intensity calibration is estimated to





be less than 3 % (2σ). Equating the error, μPNNL, with the 1σ value, i.e. 1.5 %, and assuming
that the systematic errors for all the quantities are uncorrelated, the overall systematic error in
the dataset can be given by:

$$\mu_{\text{systematic}}^2 = \mu_{\text{PNNL}}^2 + \mu_{\text{T}}^2 + \mu_{\text{P}}^2 + \mu_{\text{phot}}^2.$$                    (2)

Note that using PNNL spectra for intensity calibration effectively nullifies the errors in the
trichlorofluoromethane partial pressures and cell pathlength, so these do not have to be
included in Eq. 2.  According to Eq. 2, the systematic error contribution, μsystematic, to the new
cross sections is ~3% (1σ).


**4. Comparison between absorption cross-section datasets**

In this section the new dataset presented in this work is compared with the older

Varanasi dataset.  The comparison focuses on their wavenumber scales, integrated band
strengths, artefacts such as channel fringing, signal-to-noise ratios, spectral resolution, and
PT coverage.  In addition, the new dataset includes the weak combination band, $\nu_2 + \nu_5$, not
present in the Varanasi measurements.  This new dataset will be made available to the
community via the HITRAN and GEISA databases, but in the meantime is available
electronically from the author.

**4.1. Wavenumber scale**

It is likely that the wavenumber scale for the Varanasi dataset was never calibrated;

this has been observed in a number of recent studies for other halogenated species in which
new datasets have been compared with older Varanasi datasets, e.g. HFC-134a (Harrison,
2015a), CFC-12 (Harrison, 2015b), and HCFC-22 (Harrison, 2016).  As explained earlier, the
absolute accuracy of the wavenumber scale for the new dataset lies between 0.001 and
0.0001 cm$^{-1}$.  In comparison, the $\nu_4$ band in the Varanasi cross sections is shifted too low in
wavenumber; this shift varies between cross sections, e.g. by ~ 0.002 cm$^{-1}$ (a correction
factor of ~ 1.000002) for the 190 K / 7.5 Torr $\nu_1$ Varanasi measurement and by ~ 0.007 cm$^{-1}$
(a correction factor of ~ 1.000007) for 216.1 K / 100.0 Torr $\nu_1$.

**4.2. Integrated band strengths**



Integrated band strengths for the Varanasi cross sections have been calculated over

the spectral ranges of the cross-section files, 810 – 880 and 1050 – 1120 cm$^{-1}$, covering the $v_4$
and $v_1$ bands respectively, and compared with those for the new absorption cross sections
calculated over the same ranges; plots of integrated band strength against temperature for
each dataset and wavenumber range can be found in Figure 3. At each temperature the
Varanasi integrated band strengths display a small spread in values, most notably for the $v_4$
band, however there is no evidence for any temperature dependence, as expected. The small
spread in values is likely due to inconsistencies in the baselines for the Varanasi cross
sections. Unfortunately, the wavenumber ranges do not extend far enough to obtain an
unambiguous measure of the baseline position, and the cross sections in the HITRAN and
GEISA databases have had all negative cross section values set to zero, which has the effect
of adjusting the baseline positions by a small amount near the band wings.

**4.3. Channel fringes**

Most of the absorption cross sections in the Varanasi CFC-11 dataset contain

noticeable channel fringes above the noise level (refer to Figure 4 for an example of this); in
transmittance these would equate to peak-to-peak amplitudes as high as ~2–3 %. For the
measurements described in the present work, wedged cell windows were used to avoid
channel fringes by preventing reflections from components in the optical path of the
spectrometer.

**4.4. Signal-to-noise ratios (SNRs)**

The SNRs of the transmittance spectra measured in the present work have been

calculated using Bruker's OPUS software at ~ 990 cm$^{-1}$ where the transmittance is close to 1;
the values range from 2600 to 4700 (rms), equivalent to percentage transmittances between
0.04 and 0.02 %. A direct comparison with the Varanasi dataset, however, is not possible
without the original transmittance spectra or, at the very least, information on the
experimental mixing ratios. Further complicating issues, the Varanasi cross sections are
missing negative values near the baselines (refer to Section 4.2) and many have channel
fringes superimposed. However, it is apparent from a direct inspection that the new cross
sections have improved SNR, in some cases substantially so.

**4.5. Spectral resolution**



All spectra used to create the Varanasi cross-section dataset were either recorded at
0.01 (for sample mixtures of 75 Torr and below) or 0.03 cm$^{-1}$ spectral resolution (defined as
0.9/MOPD). In the present work 0.01 cm$^{-1}$ resolution was used for mixtures below 10 Torr,
0.03 cm$^{-1}$ for 300 Torr and above, and 0.015 and 0.0225 cm$^{-1}$ for intermediate pressures. The
spectra recorded at 191.6 K and 98.14 / 200.0 Torr were mistakenly recorded at spectral
resolutions of 0.0225 / 0.0300 cm$^{-1}$ instead of the planned 0.015 / 0.0225 cm$^{-1}$. However,
careful inspection indicated that there was no under-resolving of spectral features for these
two measurements. Overall, the dataset comparison indicates that the spectral resolutions
chosen for the Varanasi measurements were suitable.

**4.6. Pressure-temperature coverage**
An absorption cross-section dataset used in remote sensing should cover all possible
combinations of pressure and temperature appropriate for the region of the atmosphere being
observed; in this case the focus is on the mid-troposphere (~ 5 km) up to the stratosphere. It
is preferable to utilise an interpolation scheme in forward model calculations, rather than to
extrapolate beyond the temperatures and pressures represented within the dataset. Figure 5
provides a graphical representation of the PT combinations for both datasets, illustrating the
improved PT coverage (30 PT combinations in total) relative to the Varanasi dataset.


**5. Conclusions**
New high-resolution IR absorption cross sections for air-broadened
trichlorofluoromethane (CFC-11) have been determined over the spectral range 710 – 1290
cm$^{-1}$, with an estimated systematic uncertainty of ~ 3 %. Spectra were recorded at
resolutions between 0.01 and 0.03 cm$^{-1}$ (calculated as 0.9/MOPD) over a range of
atmospherically relevant temperatures and pressures (7.5 – 760 Torr and 192 – 293 K).
These new absorption cross sections improve upon those currently available in the HITRAN
and GEISA databases. In particular, they cover a wider range of pressures and temperatures,
they have a more accurately calibrated wavenumber scale, they have more consistent
integrated band intensities, they do not display any channel fringing, they have improved
SNR, and additionally they cover the weak combination band, $\nu_2 + \nu_5$.


**Acknowledgements**



The author wishes to thank the National Centre for Earth Observation (NCEO),
funded by the UK Natural Environment Research Council (NERC), for funding this work, as
well as R.G. Williams and R.A. McPheat for providing technical support during the
measurements.

**Figure Captions**


Figure 1.  The absorption cross section of trichlorofluoromethane / dry synthetic air at 191.7
K and 7.535 Torr (this work), with vibrational band assignments for the main band systems
in the 710 – 1290 cm$^{-1}$ spectral region.

Figure 2.  The new absorption cross sections of trichlorofluoromethane / dry synthetic air at a
total pressure of ~ 200.0 Torr over a range of temperatures (191.6, 202.4, 216.6, 232.6,
252.5, and 273.8 K).  The observed narrowing of the $\nu_4$ band as the temperature decreases is
due to the decline in Boltzmann populations of the upper rovibrational levels of the ground
state.

Figure 3.  Integrated band strength as a function of temperature for the new and Varanasi
cross-section datasets over the wavenumber ranges 810 – 880 and 1050 – 1120 cm$^{-1}$.

Figure 4.  The Varanasi absorption cross section of trichlorofluoromethane / dry synthetic air
at 232.7 K and 250.0 Torr (top), converted to transmittance by assuming similar sample
conditions to those in the present study.  The resulting spectrum has been wavenumber
calibrated and divided by a transmittance spectrum interpolated between the new measured
spectra at 232.6 K and 201.0 / 399.8 Torr.  Apart from the small difference arising from the
mis-match in sample conditions, the resulting ratio (bottom) indicates the presence of channel
fringes above the noise level.

Figure 5.  A graphical representation of the PT coverage for both the new and Varanasi
datasets.

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

**Tables**

Table 1: Summary of limb sounders past and present capable of measuring CFC-11.

| Instrument | Platform | |
|---|---|---|
| ATMOS (Atmospheric Trace MOlecule Spectroscopy) | Space shuttle | Chang et al., 1996; Irion et al., 2002 |
| CIRRIS 1A (Cryogenic InfraRed Radiance Instrumentation for Shuttle) | Space shuttle | Bingham et al., 1997 |
| CRISTA (CRyogenic Infrared Spectrometers and Telescopes for the Atmosphere) | Space shuttle | Offermann et al., 1999 |
| CLAES (Cryogenic Limb Array Etalon Spectrometer) | UARS (Upper Atmosphere Research Satellite) | Roche, et al., 1993 |
| ILAS (Improved Limb Atmospheric Spectrometer) | ADEOS (ADvanced Earth Observing Satellite) | Yokota, et al., 2002 |
| ILAS II | ADEOS II | Wetzel et al., 2006, |
| HIRDLS (HIgh Resolution Dynamics Limb Sounder) | Aura | Hoffmann et al., 2014 |
| MIPAS (Michelson Interferometer for Passive Atmospheric Sounding) | ENVISAT (ENVIronmental SATellite) | e.g. Hoffmann et al., 2005; Dinelli et al., 2010; Kellmann et al., 2012 |
| ACE-FTS (Atmospheric Chemistry Experiment – Fourier transform spectrometer) | SCISAT | Brown et al., 2011 |








Table 2: FTS parameters, sample conditions, and cell configuration for all measurements

| | |
|---|---|
| Spectrometer | Bruker Optics IFS 125HR |
| Mid-IR source | Globar |
| Detector | Mercury cadmium telluride (MCT) D313 [a] |
| Beam splitter | Potassium bromide (KBr) |
| Optical filter | ~700–1400 $cm^{-1}$ bandpass |
| Spectral resolution | 0.01 to 0.03 $cm^{-1}$ |
| Aperture size | 3.15 mm |
| Apodisation function | Boxcar |
| Phase correction | Mertz |
| $CCl_3F$ (Supelco) | 99.9% purity, natural-abundance isotopic mixture; freeze-pump-thaw purified multiple times prior to use |
| Air zero (BOC Gases) | total hydrocarbons < 3 ppm, $H_2O$ < 2 ppm, CO2 < 1 ppm, CO < 1 ppm; used 'as is' |
| Cell pathlength | 26 cm |
| Cell windows | Potassium bromide (KBr) (wedged) |
| Pressure gauges | 3 MKS-690A Baratrons (1, 10 & 1000 Torr) (±0.05% accuracy) |
| Refrigeration | Julabo F95-SL Ultra-Low Refrigerated Circulator (with ethanol) |
| Thermometry | 4 PRTs, Labfacility IEC 751 Class A |
| Wavenumber calibration | $N_2O$ |

[a]Due to the non-linear response of MCT detectors to the detected radiation, all interferograms
were Fourier transformed using Bruker's OPUS software with a non-linearity correction
applied.






Table 3: Summary of the sample conditions for all measurements.

| Temperature (K) | Initial $CCl_3F$ Pressure (Torr)[a] | Total Pressure (Torr) | Spectral resolution (cm$^{-1}$)[b] |
|---|---|---|---|
| 191.7 ± 0.8 | 0.266 | 7.535 ± 0.035 | 0.0100 |
| 191.5 ± 0.8 | 0.302 | 49.83 ± 0.13 | 0.0150 |
| 191.6 ± 0.8 | 0.302 | 98.14 ± 0.68 | 0.0225 |
| 191.6 ± 0.8 | 0.266 | 200.0 ± 0.3 | 0.0300 |
| 202.3 ± 0.5 | 0.319 | 7.508 ± 0.006 | 0.0100 |
| 202.4 ± 0.5 | 0.309 | 50.28 ± 0.13 | 0.0150 |
| 202.3 ± 0.5 | 0.318 | 99.85 ± 0.30 | 0.0150 |
| 202.4 ± 0.5 | 0.309 | 200.4 ± 0.2 | 0.0225 |
| 202.3 ± 0.5 | 0.318 | 301.6 ± 0.3 | 0.0300 |
| 216.7 ± 0.5 | 0.347 | 7.496 ± 0.018 | 0.0100 |
| 216.7 ± 0.5 | 0.358 | 49.93 ± 0.09 | 0.0150 |
| 216.7 ± 0.5 | 0.357 | 99.94 ± 0.07 | 0.0150 |
| 216.6 ± 0.5 | 0.375 | 201.0 ± 0.2 | 0.0225 |
| 216.7 ± 0.5 | 0.383 | 360.4 ± 0.3 | 0.0300 |
| 232.6 ± 0.4 | 0.407 | 7.500 ± 0.020 | 0.0100 |
| 232.6 ± 0.4 | 0.395 | 49.80 ± 0.15 | 0.0150 |
| 232.6 ± 0.4 | 0.544 | 99.67 ± 0.16 | 0.0150 |
| 232.6 ± 0.4 | 0.417 | 201.0 ± 0.1 | 0.0225 |
| 232.6 ± 0.4 | 0.413 | 399.8 ± 0.3 | 0.0300 |
| 252.5 ± 0.2 | 0.503 | 7.477 ± 0.003 | 0.0100 |
| 252.5 ± 0.2 | 0.486 | 50.06 ± 0.05 | 0.0150 |
| 252.5 ± 0.2 | 0.516 | 200.9 ± 0.1 | 0.0225 |
| 252.5 ± 0.2 | 0.544 | 399.9 ± 0.2 | 0.0300 |
| 252.5 ± 0.2 | 0.607 | 600.2 ± 0.3 | 0.0300 |
| 273.9 ± 0.2 | 0.475 | 7.501 ± 0.001 | 0.0100 |
| 273.8 ± 0.2 | 0.613 | 201.6 ± 0.1 | 0.0225 |
| 273.8 ± 0.2 | 0.598 | 355.8 ± 0.1 | 0.0300 |
| 273.8 ± 0.2 | 0.607 | 760.1 ± 0.2 | 0.0300 |
| 293.1 ± 0.1 | 0.548 | 355.8 ± 0.1 | 0.0300 |
| 293.0 ± 0.1 | 0.566 | 760.0 ± 0.1 | 0.0300 |

[a] MKS-690A Baratron readings are accurate to ± 0.05%.
[b] Using the Bruker definition of 0.9/MOPD.

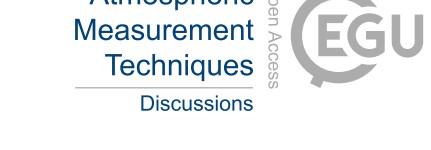

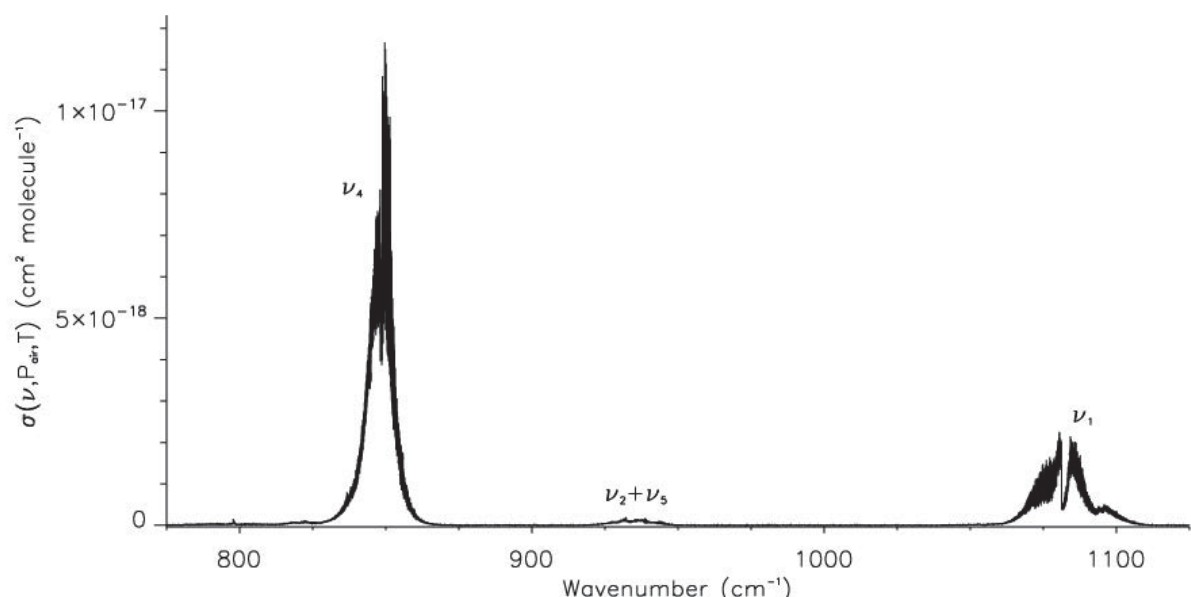




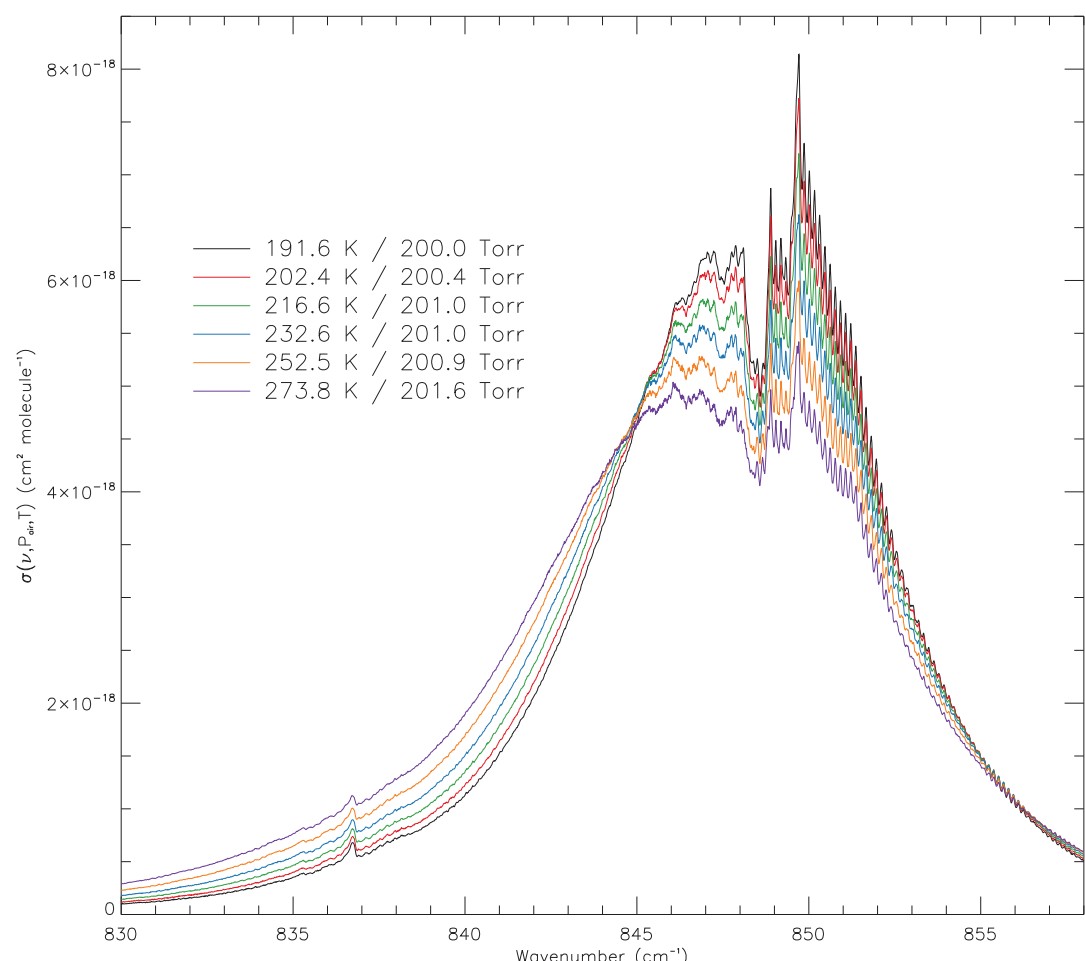





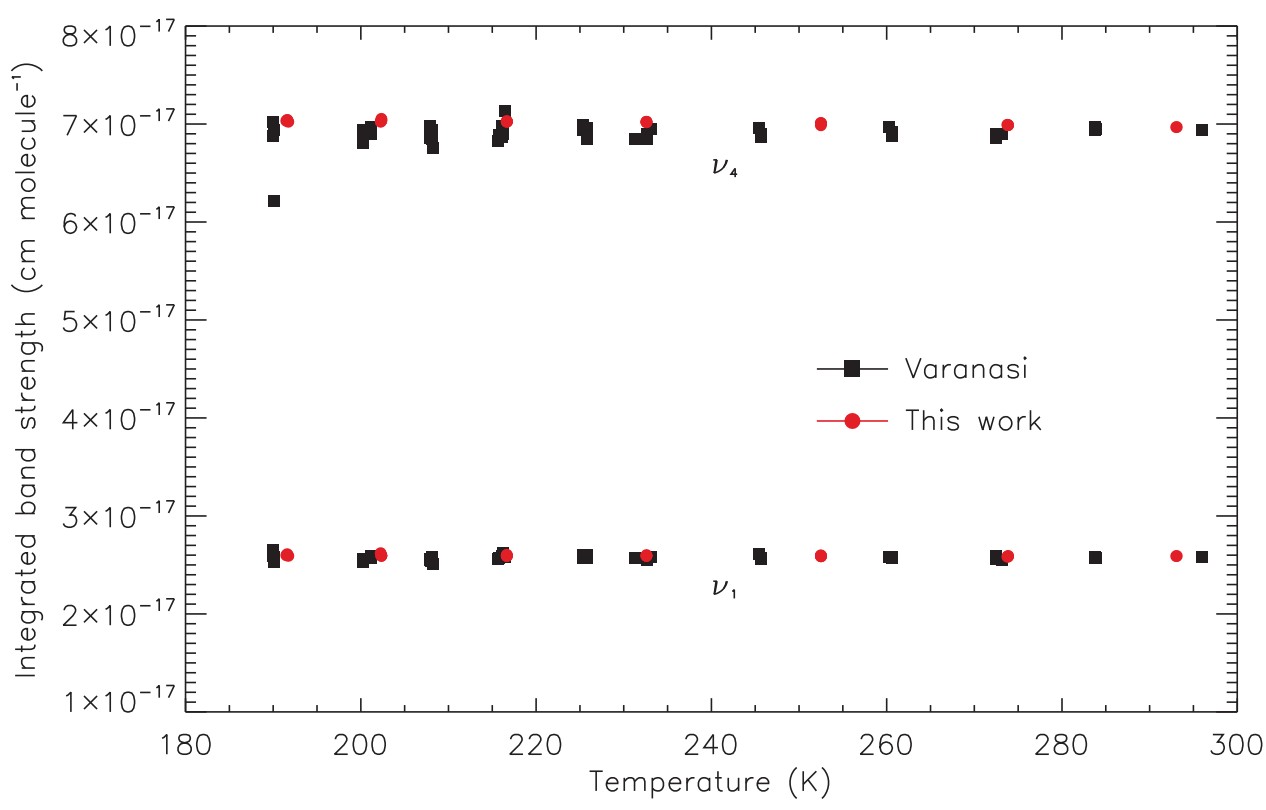



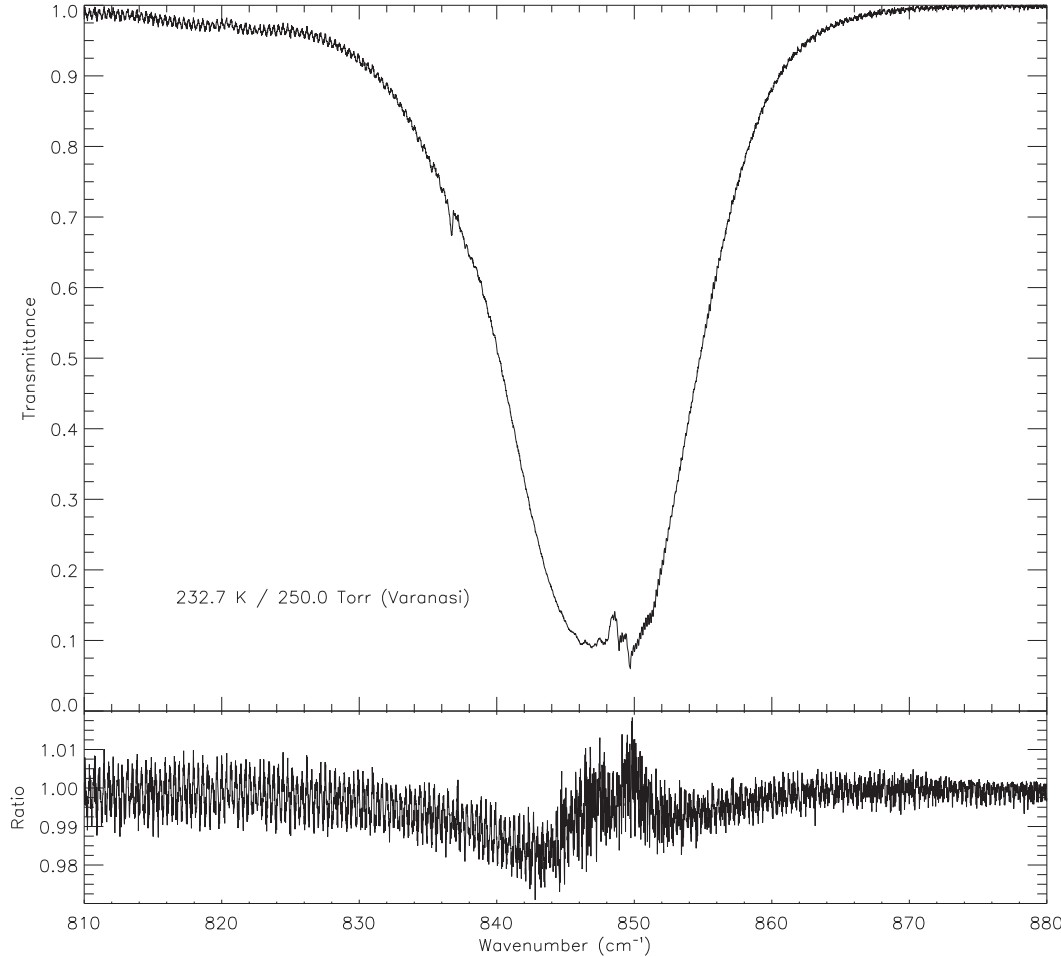





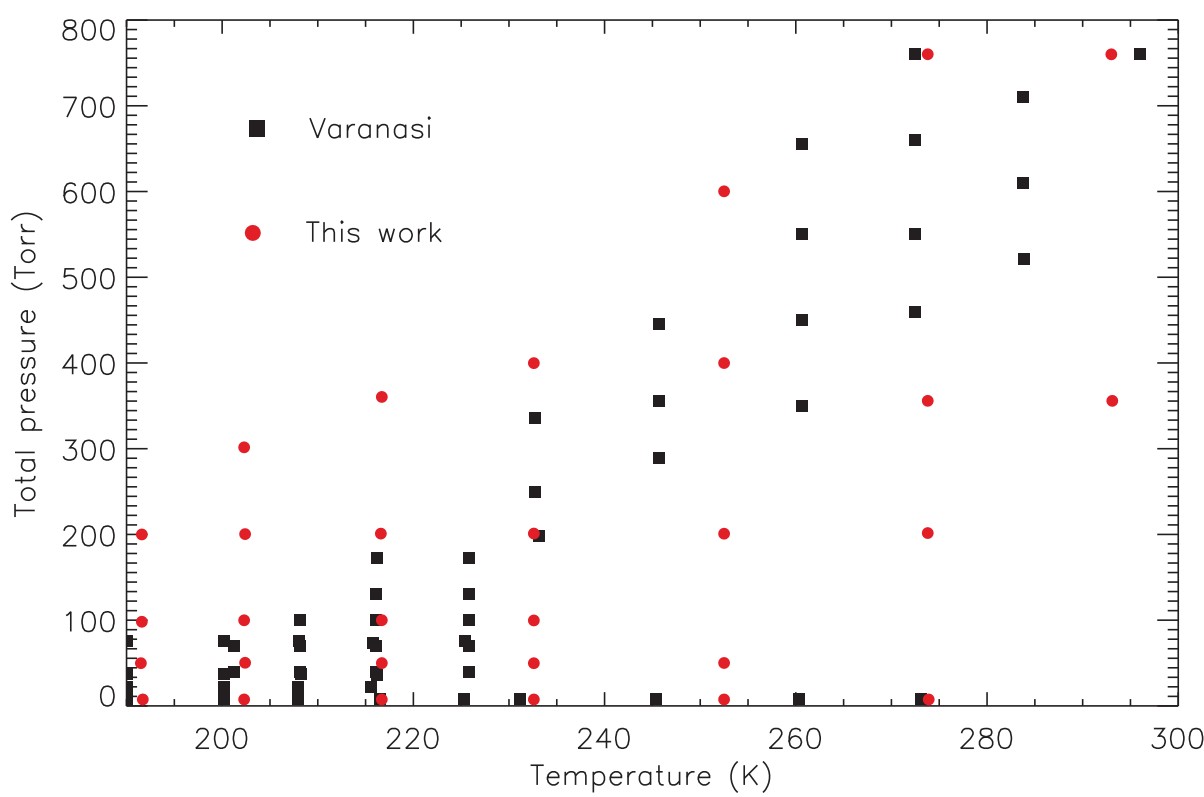