# Peer review of "New and improved infrared absorption cross sections for trichlorofluoromethane (CFC-11)"

_Atmospheric Measurement Techniques, 2018_

## Referee Comment (RC1) · Anonymous Referee #1 · 24 May 2018

General comments

In this paper, Harrison presents a new data set of absorption cross sections for trichlorofluoromethane (CFCl3 or CFC-11). The cross sections have been measured for about 30 pressure/temperature combinations, using an experimental setup and methodology introduced earlier by the same author. Overall, the paper is well written and concise. It fits in the scope of AMT and I would recommend it for publication, subject to a few specific comments listed below.

Specific comments

l52-53: It might be good to add a few words on how the new data set improves upon

the existing Varanasi data set in the abstract.

l62-65: Add a reference for the polar ozone chemistry, e.g., Solomon (1999)?

Solomon, S. (1999), Stratospheric ozone depletion: A review of concepts and history, Rev. Geophys., 37(3), 275–316, doi: 10.1029/1999RG900008.

l94: Add references for the GEISA and HITRAN databases?

l144-151: I have a question regarding the measurements which mostly arises out of my curiosity, but perhaps other readers might also be interested: How long does it actually take to make those measurements of the absorption cross sections? Is this a piece of work completed within a few hours or days? Could you easily add more p/T combinations?

l197-198: It is stated that the total systematic error of the new cross sections is ∼3%. Is this sufficient to improve retrievals for the satellite instruments? How does it compare to the Varanasi data?

l206-208: This is just one sentence, but it may go into a separate "data availability" section, following AMT author guidelines?

l216-219: Not sure if those tiny relative correction factors (1.000002 ... 1.000007) really need to be reported in addition to the absolute wavenumber shifts?

l251-252: You say it is difficult, but perhaps you could still try to show an illustrative example comparing the SNRs from your data set and the Varanasi data set? This could help demonstrate that the new data set is improving upon the existing one.

l270-272: The new data set is improving the p/T coverage, but the sampling density actually seems to be lower (fewer data points in your data set). Do you consider this lower sampling density in p/T space to be negligible, as there might potentially be low variability in the data? It would be good to show climatological p/T profiles in Fig. 5 to illustrate that your data set covers atmospheric variability.

This might be a source of climatological data (in Leicester): Remedios, J. J., Leigh, R. J., Waterfall, A. M., Moore, D. P., Sembhi, H., Parkes, I., Greenhough, J., Chipperfield, M. P., and Hauglustaine, D.: MIPAS reference atmospheres and comparisons to V4.61/V4.62 MIPAS level 2 geophysical data sets, Atmos. Chem. Phys. Discuss., 7, 9973-10017, https://doi.org/10.5194/acpd-7-9973-2007, 2007.

Table 1: This is a nice overview of CFC-11 measurements from space. You might consider adding the time frame of the measurements, e.g., 2002-2012 for MIPAS, 2005-2008 for HIRDLS, etc. and add "References" as header for the third column of the table.

Technical corrections

l84: "very many" -> "many" ?
* * *

---

## Referee Comment (RC2) · G. Toon (Referee) · 23 Jul 2018

**Review of "New and improved infrared absorption cross sections for trichlorofluoromethane (CFC-11), Jeremy Harrison, AMT-2018-51**

This review was written in March 2018 and refers to the original submitted document. So the line numbers cited below may have changed and some of the comments may no longer apply.

**Overview.**

This paper reports new infrared lab measurements of CFC-11, an important GHG and ODS. These new measurements represent an important contribution to remote sensing of CFC-11 and so a paper describing them is certainly merited. That said, the paper has certain deficiencies that need correction. The author claims that the new measurements are better than previous lab measurements, i.e. Li and Varanasi, which is probably true, but the supporting evidence that is presented is not compelling. There are also aspects of the new measurements that are, in my opinion, inferior to previous measurements. This needs more discussion. Also, the error budget seems to omit some potentially important terms (random error, detector non-linearity, intensity calibration).

**Specific Comments.**

1) The new cross-sections rely on PNNL data for absolute intensity calibration, rather than by independently measuring the amount of gas in the cell. The author states that this "is necessary to counter problems with trichlorofluoromethane adsorption in the vacuum line and on the cell walls, resulting in its partial pressure during each measurement differing from the initial, measured value". The author needs to explain why this "adsorption" wasn't a problem for PNNL or for Li and Varanasi [1994].

2) The PNNL measurements cover a rather high temperature range (278-323K). The present work covers 191-293K, with only 2/30 spectra exceeding 274K. Despite this minimal overlap in temperature space, the author nevertheless uses the PNNL spectra to calibrate their cross-sections, implicitly assuming that the band intensities are T-independent. Please discuss the validity of this assumption and its likely impact on the error budget.

On a similar topic, lines 263-265 state: "The Varanasi integrated band strengths at each temperature display a small spread in values, most notably for the ν4 band, however there is no evidence for any temperature dependence, as expected." Why is this expected? [I'm not saying that the statement is incorrect; merely that slightly more explanation is needed]

3) Section 3.3. The author claims that: "random errors in y (transmittance) cannot be determined since only one spectrum is recorded at each PT combination". And yet, in the conclusions (lines 319-320), the author asserts that the SNR of his new spectra is superior to Li and Varanasi's. This latter statement implies that the author can, in fact, estimate the SNR of his spectra, in which case it can be included as a random term in the error budget.

4) Section 3.3. The author claims a total systematic error of ~3%. This includes "photometric uncertainty" which he doesn't define. Please elaborate.

5) I would guess that an important error in this type of work is zero-level offsets due to detection non-linearity. The author states that the Bruker OPUS software was used to correct for detector non-linearity. While this may reduce the zero level offsets by an order of magnitude, it won't be perfect. So the error analysis must still include an estimate of the effect of residual zero-level offset. For example, If the spectra have a residual zero-level offset that is just 0.3% of the continuum, and if the gas transmittance falls to 6% in the band center, as depicted in fig. 4, then the resulting error in the cross sections will be 0.003/0.060 = 5% at band center and will dominate the error budget.

6) Table 1 provides no information on the length of the cell, although the abstract says 26 cm. This needs to be included.

7) The new measurements seem to have fewer spectra than Varanasi's with larger temperature gaps. I counted 55 different points in figure 5 representing Varanasi's measurements versus 30 for the new work. Please discuss the reasoning behind this coarser temperature sampling and its implications for remote sensing.

8) The author asserts that his new cross-sections are better than previous ones due to the wider range of T/P. But when I look at fig. 5 the only places where the P/T coverage is extended by the new measurements is near 285 K/300 Torr and around 200 K/300 Torr, conditions that rarely happen in Earth's atmosphere. And the new measurements have a huge "hole" around 275±20 K and 560±150 Torr, a very common atmospheric condition. So in terms of PT coverage, the new measurements seem worse than those of Li and Varanasi. Perhaps the new measurements are intended to complement previous ones, rather than be a stand-alone data-base. But there is no statement of this intention. Even more disappointing is the continued absence of lab measurements covering 240 K/750 Torr, conditions that happen every winter over vast regions of the globe (Canada, Russia, Arctic, Antarctic).

The author should add standard temperature profiles, such as the three below (found on internet), to figure 5, after converting altitude to pressure. Readers will then be able to better judge the benefits of the new extended P/T coverage.

[Figure]

9) Firstly, since fig. 4 has two panels, the caption should describe each panel separately, not leave it to the reader to figure it out. I *think* that the upper panel is a Varanasi transmittance spectrum, and the lower panel is the ratio of Varanasi/Harrison transmittances. Unfortunately, you can't really tell whether the systematic differences in the lower panel are the due to intrinsic differences in the cross-sections, or the large

pressure-interpolation (across 200-400 Torr) performed to the Harrison spectra to match the Varanasi pressure of 250 Torr.

Secondly, it seems a very odd decision to use the 250 Torr Varanasi spectrum, requiring P-interpolation, when there is already a Varanasi spectrum at 200 Torr that would have avoided interpolation. The 250 Torr, 233 K Varanasi and Harrison points overlap in fig.5. Please explain why you went to the trouble of performing a seemingly unnecessary P-interpolation.

10) Line 65: Insert "impending" before "environmental disaster". It would be an exaggeration to represent the springtime $O_3$ loss over Antarctica as an "environmental disaster". It might have become one eventually, but disaster was averted by the Montreal protocol.

11) The author repeatedly asserts that it is a "difficult" or "virtually impossible" task to "derive" spectroscopic line parameters for large molecules like CFC-11. I believe that the author is referring to a quantum-mechanically-based derivation since it is fairly straight-forward to derive an empirical "pseudo" line list for CFC-11 from lab measurements. So the author should elaborate on what he means by "derive".

12) Lines 117-118 & 121-127: Discussion of point groups and symmetry classes in section 2 should be deleted or moved into an appendix. This won't hurt because there is nothing in the subsequent paper that relates to these things anyway. The paper has been submitted to AMT and so very few readers will be familiar with these spectroscopic concepts. If the author wants to talk about quantum mechanics, he should have submitted the paper elsewhere (e.g., J. Mol. Spec.).

13) I'm not sure what fig. 3 is really telling me. The new integrated band strengths are very similar to Varanasi's values. But the new band strengths have been calibrated into agreement with PNNL anyway, so fig.3 seems to show that Varanasi agrees with PNNL. Why are the PNNL band strengths not included in this figure?

14) Lines 131-136: The discussion here has much in common with lines 96-100. I suggest removing one or the other to avoid repetition.

15) Line 208: Units should be written as: $cm^{-1}/(molecules.cm^{-2})$ as in the latest HITRAN papers. [Yes, I realize that the $cm^{-1}$ in the numerator can be cancelled, but to do so is anti-intuitive.]

16) Line 217: I don't understand the use of "x" to denote wavenumber, when "$v$" has already been defined for this purpose, e.g. on lines 206 and 208.

17) Line 275: claims Varanasi's channel fringes are as high at 2-3%. But I don't see anything over 2% in fig.4.

18) Line 292: Does " In this work..." refer to Li and Varanasi or to Harrison [2018]? If the former, use " In that work...". If the latter, use " In the present work...".

19) Table 2: Please align the decimal points in the third column.

---

## Author Response (AR1)

**Response to Reviewers of "New and improved infrared absorption cross sections for trichlorofluoromethane (CFC-11)"**

Comments are reproduced below in bold text, followed by my response.

Reviewer #1:

**Harrison presents a new data set of absorption cross sections for trichlorofluoromethane (CFCl3 or CFC-11). The cross sections have been measured for about 30 pressure/temperature combinations, using an experimental setup and methodology introduced earlier by the same author. Overall, the paper is well written and concise. It fits in the scope of AMT and I would recommend it for publication, subject to a few specific comments listed below.**

**Specific comments**
**l52-53: It might be good to add a few words on how the new data set improves upon the existing Varanasi data set in the abstract.**
This has been done.

**l62-65: Add a reference for the polar ozone chemistry, e.g., Solomon (1999)? Solomon, S. (1999), Stratospheric ozone depletion: A review of concepts and history, Rev. Geophys., 37(3), 275–316, doi: 10.1029/1999RG900008.**
This was done.

**l94: Add references for the GEISA and HITRAN databases?**
This has been done.

**l144-151: I have a question regarding the measurements which mostly arises out of my curiosity, but perhaps other readers might also be interested: How long does it actually take to make those measurements of the absorption cross sections? Is this a piece of work completed within a few hours or days? Could you easily add more p/T combinations?**
In total the measurements took about a week, which included a considerable amount of out-of-hours work. As we pay to use the facility, time is money so the measured PT combinations need to be carefully considered.

**l197-198: It is stated that the total systematic error of the new cross sections is 3%. Is this sufficient to improve retrievals for the satellite instruments? How does it compare to the Varanasi data?**
It is stated in Li & Varanasi (1994) that the actual uncertainty of their cross sections is 2%, however given the various problems identified in the present manuscript, the true uncertainty must be larger. This has been added to the manuscript. The uncertainty of the new measurements is 3 %. I believe the new data will provide a more accurate basis for retrieving CFC-11, however it must be realised that there are additional, and usually larger, sources of uncertainty in satellite measurements.

**l206-208: This is just one sentence, but it may go into a separate "data availability" section, following AMT author guidelines?**
I have added a new data availability section just before the acknowledgements.

**l216-219: Not sure if those tiny relative correction factors (1.000002 ... 1.000007) really need to be reported in addition to the absolute wavenumber shifts?**
I report these tiny shifts because the calibration factors are multiplicative, i.e. absolute shifts will differ between bands.

**l251-252: You say it is difficult, but perhaps you could still try to show an illustrative example comparing the SNRs from your data set and the Varanasi data set? This could help demonstrate that the new data set is improving upon the existing one.**
I have added a new figure (number 5) to the manuscript which illustrates the difference in SNR near the baseline for measurements at ~ 16.5 K and 7.5 Torr.

**l270-272: The new data set is improving the p/T coverage, but the sampling density actually seems to be lower (fewer data points in your data set). Do you consider this lower sampling density in p/T space to be negligible, as there might potentially be low variability in the data?**
The IR bands of CFC-11 are congested and there are no strong, sharp features. This means that there isn't a large amount of variation between cross sections and a lower sampling density in PT space is perfectly fine for remote sensing. I have added a point to this effect in the text.

**It would be good to show climatological p/T profiles in Fig. 5 to illustrate that your data set covers atmospheric variability.**
I understand the reasoning behind this request, however the climatological profiles only represent "averages" of the atmospheric variability, not actual variability. In fact, the original Li & Varanasi CFC-11 paper does include such a figure, and their PT combinations do cover these atmospheric profiles. The new data, therefore, will also cover these profiles. The PT coverage in this work is chosen to cover the range of P and T from ACE-FTS v3.0 data. I have added this point to the manuscript.

**Table 1: This is a nice overview of CFC-11 measurements from space. You might consider adding the time frame of the measurements, e.g., 2002-2012 for MIPAS, 2005-2008 for HIRDLS, etc. and add "References" as header for the third column of the table.**
Yes, this has been done.

**Technical corrections**
**l84: "very many" -> "many" ?**
"Very many" is perfectly acceptable English.

Reviewer #2:

**This review was written in March 2018 and refers to the original submitted document. So the line numbers cited below may have changed and some of the comments may no longer apply.**
Technical corrections (points 6, 9 10, 14, 15, 16, 17, 18, 19) were answered when producing the discussion manuscript.

**1) The new cross-sections rely on PNNL data for absolute intensity calibration, rather than by independently measuring the amount of gas in the cell. The author states that**

this "**is necessary to counter problems with trichlorofluoromethane adsorption in the vacuum line and on the cell walls, resulting in its partial pressure during each measurement differing from the initial, measured value**". **The author needs to explain why this "adsorption" wasn't a problem for PNNL or for Li and Varanasi [1994].**

The PNNL sample cell and gas manifold are electro-polished and gold-plated to minimise adsorption. It isn't clear from the literature whether the Varanasi cell has any special features to minimise adsorption.

**2) The PNNL measurements cover a rather high temperature range (278-323K). The present work covers 191-293K, with only 2/30 spectra exceeding 274K. Despite this minimal overlap in temperature space, the author nevertheless uses the PNNL spectra to calibrate their cross-sections, implicitly assuming that the band intensities are T independent. Please discuss the validity of this assumption and its likely impact on the error budget.**

This assumption has been explained in a previous publication, and the reader is referred to this in the text:

"This intensity calibration procedure … furthermore assumes that the integrated intensity over each band system is independent of temperature. The reader is referred to Harrison et al. (2010) for a more complete explanation of the underlying assumption, and references cited within Harrison (2015a, 2015b, and 2016) for details on previous successful uses of this approach."

**On a similar topic, lines 263-265 state: "The Varanasi integrated band strengths at each temperature display a small spread in values, most notably for the v4 band, however there is no evidence for any temperature dependence, as expected." Why is this expected? [I'm not saying that the statement is incorrect; merely that slightly more explanation is needed]**

The assumption made above is that the integrated intensity over each band system is independent of temperature. The Varanasi band strengths indicate the validity of this assumption. I have provided additional clarification in the text.

**3) Section 3.3. The author claims that: "random errors in y (transmittance) cannot be determined since only one spectrum is recorded at each PT combination". And yet, in the conclusions (lines 319-320), the author asserts that the SNR of his new spectra is superior to Li and Varanasi's. This latter statement implies that the author can, in fact, estimate the SNR of his spectra, in which case it can be included as a random term in the error budget.**

Section 4.4 provides additional information on the SNR comparison, not just the conclusion. The SNR can be estimated near the baseline, between bands – the values, which were already included in the manuscript, range from 2600 to 4700 (rms), equivalent to percentage transmittances between 0.04 and 0.02 %. This contribution is too small to have any noticeable effect on the overall error budget. The sentence quoted above is intended to refer to random uncertainties in the measurements over the full range of wavenumbers, not just at the baseline. This point has been clarified in the text.

**4) Section 3.3. The author claims a total systematic error of ~3%. This includes "photometric uncertainty" which he doesn't define. Please elaborate.**

Photometric uncertainty is associated with the detection of radiation by the MCT detector and any uncertainties due to the non-linearity correction. This point has been added to the text.

**5) I would guess that an important error in this type of work is zero-level offsets due to detection non-linearity. The author states that the Bruker OPUS software was used to correct for detector non-linearity. While this may reduce the zero level offsets by an order of magnitude, it won't be perfect. So the error analysis must still include an estimate of the effect of residual zero-level offset. For example, If the spectra have a residual zero level offset that is just 0.3% of the continuum, and if the gas transmittance falls to 6% in the band center, as depicted in fig. 4, then the resulting error in the cross sections will be 0.003/0.060 = 5% at band center and will dominate the error budget.**

I agree that the Bruker correction isn't perfect, however checks are performed during the experimental campaigns by running one of the PT measurements for less absorber amount. These comparisons indicate that any systematic error is small, certainly less than 5 %.  I assign this an upper limit of 2 % to the error budget.

**6) Table 1 provides no information on the length of the cell, although the abstract says 26 cm.  This needs to be included.**

This has already been corrected.

**7) The new measurements seem to have fewer spectra than Varanasi's with larger temperature gaps. I counted 55 different points in figure 5 representing Varanasi's measurements versus 30 for the new work. Please discuss the reasoning behind this coarser temperature sampling and its implications for remote sensing.**

This was dealt with in comments to reviewer 1, above.

**8) The author asserts that his new cross-sections are better than previous ones due to the wider range of T/P. But when I look at fig. 5 the only places where the P/T coverage is extended by the new measurements is near 285 K/300 Torr and around 200 K/300 Torr, conditions that rarely happen in Earth's atmosphere. And the new measurements have a huge "hole" around 275±20 K and 560±150 Torr, a very common atmospheric condition.  So in terms of PT coverage, the new measurements seem worse than those of Li and Varanasi. Perhaps the new measurements are intended to complement previous ones, rather than be a stand-alone data-base. But there is no statement of this intention. Even more disappointing is the continued absence of lab measurements covering 240 K/750 Torr, conditions that happen every winter over vast regions of the globe (Canada, Russia, Arctic, Antarctic).**

The wider range of T/P is ONE of the criteria used in the comparison with Varanasi data.  As mentioned above, the PT coverage in this work was chosen to cover the range of P and T from ACE-FTS v3.0 data.  Assuming the standard four point interpolation scheme, the additional range of P and T will ensure a better coverage of the atmospheric measurements. This point has been added to the manuscript.

Note that it is specified in the text that these new measurements are in support of satellite remote sensing measurements in the limb; this rules out any atmospheric conditions below 5 km in altitude.  The points about no lab measurements covering 240 K/750 Torr and the "hole" around 275±20 K and 560±150 Torr are therefore not relevant.  Having said this, spectra around 275±20 K and 560±150 Torr are less structured, so the PT sampling density doesn't need to be as high as in the Li and Varanasi dataset.

**The author should add standard temperature profiles, such as the three below (found on internet), to figure 5, after converting altitude to pressure. Readers will then be able to better judge the benefits of the new extended P/T coverage.**

This issue was addressed in the comments made by reviewer one.

**9) Firstly, since fig. 4 has two panels, the caption should describe each panel separately, not leave it to the reader to figure it out. I \*think\* that the upper panel is a Varanasi transmittance spectrum, and the lower panel is the ratio of Varanasi/Harrison transmittances. Unfortunately, you can't really tell whether the systematic differences in the lower panel are the due to intrinsic differences in the cross--sections, or the large pressure--interpolation (across 200--400 Torr) performed to the Harrison spectra to match the Varanasi pressure of 250 Torr.**

I have decided to redo this figure completely, and have written new explanatory text.

**Secondly, it seems a very odd decision to use the 250 Torr Varanasi spectrum, requiring P-interpolation, when there is already a Varanasi spectrum at 200 Torr that would have avoided interpolation. The 250 Torr, 233 K Varanasi and Harrison points overlap in fig.5. Please explain why you went to the trouble of performing a seemingly unnecessary P- interpolation.**

I have redone this figure so an interpolation is no longer needed.

**10) Line 65: Insert "impending" before "environmental disaster". It would be an exaggeration to represent the springtime O3 loss over Antarctica as an "environmental disaster". It might have become one eventually, but disaster was averted by the Montreal protocol.**

This has already been corrected.

**11) The author repeatedly asserts that it is a "difficult" or "virtually impossible" task to "derive" spectroscopic line parameters for large molecules like CFC-11. I believe that the author is referring to a quantum-mechanically-based derivation since it is fairly straight-forward to derive an empirical "pseudo" line list for CFC-11 from lab measurements. So the author should elaborate on what he means by "derive".**

The reviewer is correct that it is straightforward to derive a pseudo-linelist. However, the term "spectroscopic line parameter" implicitly refers to lines with quantum mechanical assignments. Pseudo-lines are "effective" lines (in HITRAN-type format) calculated from a set of absorption cross sections; they are not lines in the true spectroscopic sense of the word. I do not believe this point requires further clarification.

**12) Lines 117-118 & 121-127: Discussion of point groups and symmetry classes in section 2 should be deleted or moved into an appendix. This won't hurt because there is nothing in the subsequent paper that relates to these things anyway. The paper has been submitted to AMT and so very few readers will be familiar with these spectroscopic concepts. If the author wants to talk about quantum mechanics, he should have submitted the paper elsewhere (e.g., J. Mol. Spec.).**

Any AMT reader with a good grounding in spectroscopy will understand these concepts. Quantum mechanics and symmetry are cornerstones of spectroscopy, so I don't believe the inclusion of these sentences is problematic.

**13) I'm not sure what fig. 3 is really telling me. The new integrated band strengths are very similar to Varanasi's values. But the new band strengths have been calibrated into agreement with PNNL anyway, so fig.3 seems to show that Varanasi agrees with PNNL. Why are the PNNL band strengths not included in this figure?**

Yes, the agreement between Varanasi and PNNL band strengths are very good. The PNNL band strengths are similar to the band strengths of the new measurements. These have now been added to the figure as the reviewer wishes.

**14) Lines 131---136: The discussion here has much in common with lines 96---100. I suggest removing one or the other to avoid repetition.**
This has already been corrected.

**15) Line 208: Units should be written as: cm-1/(molecules.cm-2) as in the latest HITRAN papers. [Yes, I realize that the cm-1 in the numerator can be cancelled, but to do so is anti-intuitive.]**
The conventional units for absorption cross sections, as given in the recent HITRAN 2016 paper, are $cm^2$ molecule$^{-1}$. Integration of a cross section with respect to wavenumber (cm$^{-1}$) results in an integrated band intensity with units cm molecule$^{-1}$. The units given by the reviewer above correspond to the intensity of a single spectroscopic line, however "cm molecule$^{-1}$" is more in keeping with the established convention for cross sections, even if it is counter-intuitive.

**16) Line 217: I don't understand the use of "x" to denote wavenumber, when "v" has already been defined for this purpose, e.g. on lines 206 and 208.**
In the context of this discussion, x was referring to the x-axis. This has already been corrected.

**17) Line 275: claims Varanasi's channel fringes are as high at 2-3%. But I don't see anything over 2% in fig.4.**
This is just one cross section out of 55 – the magnitude of the fringing varies between cross sections.

**18) Line 292: Does " In this work..." refer to Li and Varanasi or to Harrison [2018]? If the former, use " In that work...". If the latter, use " In the present work...".**
This has already been corrected.

**19) Table 2: Please align the decimal points in the third column.**
This is a type-setting issue for the final published version.

**New and improved infrared absorption cross sections for trichlorofluoromethane (CFC-11)**

by

Jeremy J. Harrison[1,2,3]

[1]*Department of Physics and Astronomy, University of Leicester, Leicester LE1 7RH, United Kingdom.*

[2]*National Centre for Earth Observation, University of Leicester, Leicester LE1 7RH, United Kingdom.*

[3]*Leicester Institute for Space and Earth Observation, University of Leicester, Leicester LE1 7RH, United Kingdom.*

Number of pages   = 18
Number of tables   = 3
Number of figures = 5

Address for correspondence:

        Dr. Jeremy J. Harrison
        National Centre for Earth Observation
        Department of Physics and Astronomy
        University of Leicester
        University Road
        Leicester LE1 7RH
        United Kingdom

*e-mail*:        jh592@leicester.ac.uk

**Abstract**

Trichlorofluoromethane (CFC-11), a widely used refrigerant throughout much of the twentieth century and a very potent (stratospheric) ozone depleting substance (ODS), is now banned under the Montreal Protocol. With a long atmospheric lifetime, it will only slowly degrade in the atmosphere, so monitoring its vertical concentration profile using infrared-sounding instruments, thereby validating stratospheric loss rates in atmospheric models, is of great importance; this in turn requires high quality laboratory spectroscopic data.

This work describes new high-resolution infrared absorption cross sections of trichlorofluoromethane / dry synthetic air over the spectral range 710 – 1290 cm$^{-1}$, determined from spectra recorded using a high-resolution Fourier transform spectrometer (Bruker IFS 125HR) and a 26-cm-pathlength cell. Spectra were recorded at resolutions between 0.01 and 0.03 cm$^{-1}$ (calculated as 0.9/MOPD; MOPD = maximum optical path difference) over a range of temperatures and pressures (7.5 – 760 Torr and 192 – 293 K) appropriate for atmospheric conditions. This new cross-section dataset improves upon the one currently available in the HITRAN and GEISA databases through an extension to the range of pressures and temperatures, better signal-to-noise and wavenumber calibrations, the lack of channel fringing, the better consistency in integrated band intensities, and additionally the coverage of the weak combination band $\nu2 + \nu5$.

[revised manuscript text omitted]

and 0.7 %, respectively (see Table 3). The photometric uncertainty ($\mu_{phot}$), associated with
the detection of radiation by the MCT detector and systematic error arising from the use of
Bruker's non-linearity correction for MCT detectors, is estimated to be ~2 %. The
pathlength error ($\mu_{path}$) is estimated to be negligibly small, lower than 0.1 %. According to
the PNNL metadata, the systematic error in the PNNL $CCl_3F$ spectra used for the intensity
calibration is estimated to be less than 3 % ($2\sigma$). Equating the error, $\mu_{PNNL}$, with the $1\sigma$
value, i.e. 1.5 %, and assuming that the systematic errors for all the quantities are
uncorrelated, the overall systematic error in the dataset can be given by:

$$\mu_{systematic}^2 = \mu_{PNNL}^2 + \mu_T^2 + \mu_P^2 + \mu_{phot}^2. \qquad (2)$$

Note that using PNNL spectra for intensity calibration effectively nullifies the errors in the
trichlorofluoromethane partial pressures and cell pathlength, so these do not have to be
included in Eq. 2. According to Eq. 2, the systematic error contribution, $\mu_{systematic}$, to the new
cross sections is ~3% ($1\sigma$).

**4. Comparison between absorption cross-section datasets**

In this section the new dataset presented in this work is compared with the older
Varanasi dataset, which has a stated uncertainty of 2 % (Li and Varanasi, 1994. The
comparison focuses on their wavenumber scales, integrated band strengths, artefacts such as
channel fringing, signal-to-noise ratios, spectral resolution, and PT coverage. Given the
various problems identified in sections below, the 2 % uncertainty is a significant
underestimate. In addition, the new dataset includes the weak combination band, $\nu_2 + \nu_5$, not
present in the Varanasi measurements. These new data will provide a more accurate basis for
retrieving CFC-11 from atmospheric spectra recorded in the limb.

**4.1. Wavenumber scale**

It is likely that the wavenumber scale for the Varanasi dataset was never calibrated;
this has been observed in a number of recent studies for other halogenated species in which
new datasets have been compared with older Varanasi datasets, e.g. HFC-134a (Harrison,
2015a), CFC-12 (Harrison, 2015b), and HCFC-22 (Harrison, 2016). As explained earlier, the absolute accuracy of the wavenumber scale for the new dataset lies between 0.001 and 0.0001 $cm^{-1}$. In comparison, the $\nu_4$ band in the Varanasi cross sections is shifted too low in wavenumber; this shift varies between cross sections, e.g. by $\sim 0.002$ $cm^{-1}$ (a correction factor of $\sim 1.000002$) for the 190 K / 7.5 Torr $\nu_1$ Varanasi measurement and by $\sim 0.007$ $cm^{-1}$ (a correction factor of $\sim 1.000007$) for 216.1 K / 100.0 Torr $\nu_1$.

**4.2. Integrated band strengths**

Integrated band strengths for the Varanasi cross sections have been calculated over the spectral ranges of the cross-section files, 810 – 880 and 1050 – 1120 $cm^{-1}$, covering the $\nu_4$ and $\nu_1$ bands respectively, and compared with those for the new absorption cross sections calculated over the same ranges; plots of integrated band strength against temperature for each dataset, including the PNNL spectra, and wavenumber range can be found in Figure 3. At each temperature the Varanasi integrated band strengths display a small spread in values, most notably for the $\nu_4$ band, however there is no evidence for any temperature dependence, backing up the assumption in Section 3.2 that the integrated intensity over each band system is independent of temperature. The small spread in values is likely due to inconsistencies in the baselines for the Varanasi cross sections, which are larger for the $\nu_4$ band. Additionally, according to the PNNL spectra and the new measurements, the $\nu_4$ cross section at 810 $cm^{-1}$ is non-zero due to the presence of a weak hot band. Therefore, calculating integrated band strengths for the new dataset over the 810 – 880 $cm^{-1}$ range creates a very small temperature dependence in the $\nu_4$ integrated band strengths. Unfortunately, the wavenumber ranges do not extend far enough to obtain an unambiguous measure of the baseline position for the Varanasi data, and the cross sections in the HITRAN and GEISA databases have had all negative cross section values set to zero, which has the effect of adjusting the baseline positions by a small amount near the band wings.

**4.3. Channel fringes**

[revised manuscript text omitted]